# Low Rank Approximation Lower Bounds in Row-Update Streams

**David P. Woodruff**
IBM Research Almaden
`dpwoodru@us.ibm.com`

## Abstract

We study low-rank approximation in the streaming model in which the rows of an $n \times d$ matrix $A$ are presented one at a time in an arbitrary order. At the end of the stream, the streaming algorithm should output a $k \times d$ matrix $R$ so that $\|A - AR^\dagger R\|_F^2 \leq (1+\epsilon)\|A - A_k\|_F^2$, where $A_k$ is the best rank-$k$ approximation to $A$. A deterministic streaming algorithm of Liberty (KDD, 2013), with an improved analysis of Ghashami and Phillips (SODA, 2014), provides such a streaming algorithm using $O(dk/\epsilon)$ words of space. A natural question is if smaller space is possible. We give an almost matching lower bound of $\Omega(dk/\epsilon)$ bits of space, even for randomized algorithms which succeed only with constant probability. Our lower bound matches the upper bound of Ghashami and Phillips up to the word size, improving on a simple $\Omega(dk)$ space lower bound.

## 1 Introduction

In the last decade many algorithms for numerical linear algebra problems have been proposed, often providing substantial gains over more traditional algorithms based on the singular value decomposition (SVD). Much of this work was influenced by the seminal work of Frieze, Kannan, and Vempala [8]. These include algorithms for matrix product, low rank approximation, regression, and many other problems. These algorithms are typically approximate and succeed with high probability. Moreover, they also generally only require one or a small number of passes over the data.

When the algorithm only makes a single pass over the data and uses a small amount of memory, it is typically referred to as a *streaming algorithm*. The memory restriction is especially important for large-scale data sets, e.g., matrices whose elements arrive online and/or are too large to fit in main memory. These elements may be in the form of an entry or entire row seen at a time; we refer to the former as the entry-update model and the latter as the row-update model. The row-update model often makes sense when the rows correspond to individual entities. Typically one is interested in designing robust streaming algorithms which do not need to assume a particular order of the arriving elements for their correctness. Indeed, if data is collected online, such an assumption may be unrealistic.

Muthukrishnan asked the question of determining the memory required of data stream algorithms for numerical linear algebra problems, including best rank-$k$ approximation, matrix product, eigenvalues, determinants, and inverses [18]. This question was posed again by Sarlós [21]. A number of exciting streaming algorithms now exist for matrix problems. Sarlós [21] gave 2-pass algorithms for matrix product, low rank approximation, and regression, which were sharpened by Clarkson and Woodruff [5], who also proved lower bounds in the entry-update model for a number of these problems. See also work by Andoni and Nguyen for estimating eigenvalues in a stream [2], and work in [1, 4, 6] which implicitly provides algorithms for approximate matrix product.

In this work we focus on the low rank approximation problem. In this problem we are given an $n \times d$ matrix $A$ and would like to compute a matrix $B$ of rank at most $k$ for which $\|A - B\|_F \leq$

$(1+\epsilon)\|A - A_k\|_F$. Here, for a matrix $A$, $\|A\|_F$ denotes its Frobenius norm $\sqrt{\sum_{i=1}^{n} \sum_{j=1}^{d} A_{i,j}^2}$ and $A_k$ is the best rank-$k$ approximation to $A$ in this norm given by the SVD.

Clarkson and Woodruff [5] show in the entry-update model, one can compute a factorization $B = L \cdot U \cdot R$ with $L \in \mathbb{R}^{n \times k}$, $U \in \mathbb{R}^{k \times k}$, and $R \in \mathbb{R}^{k \times d}$, with a streaming algorithm using $O(k\epsilon^{-2}(n + d/\epsilon^2)\log(nd))$ bits of space. They also show a lower bound of $\Omega(k\epsilon^{-1}(n + d)\log(nd))$ bits of space. One limitation of these bounds is that they hold only when the algorithm is required to output a factorization $L \cdot U \cdot R$. In many cases $n \gg d$, and using memory that grows linearly with $n$ (as the above lower bounds show is unavoidable) is prohibitive. As observed in previous work [9, 16], in downstream applications we are often only interested in an approximation to the top $k$ principal components, i.e., the matrix $R$ above, and so the lower bounds of Clarkson and Woodruff can be too restrictive. For example, in PCA the goal is to compute the most important directions in the row space of $A$.

By reanalyzing an algorithm of Liberty [16], Ghashami and Phillips [9] were able to overcome this restriction in the row-update model, showing that Liberty's algorithm is a streaming algorithm which finds a $k \times d$ matrix $R$ for which $\|A - AR^{\dagger}R\|_F \leq (1+\epsilon)\|A - A_k\|_F$ using only $O(dk/\epsilon)$ words of space. Here $R^{\dagger}$ is the Moore-Penrose pseudoinverse of $R$ and $R^{\dagger}R$ denotes the projection onto the row space of $R$. Importantly, this space bound no longer depends on $n$. Moreover, their algorithm is deterministic and achieves relative error. We note that Liberty's algorithm itself is similar in spirit to earlier work on incremental PCA [3, 10, 11, 15, 19], but that work missed the idea of using a Misra-Gries heavy hitters subroutine [17] which is used to bound the additive error (which was then improved to relative error by Ghashami and Phillips). It also seems possible to obtain a streaming algorithm using $O(dk(\log n)/\epsilon)$ words of space, using the coreset approach in an earlier paper by Feldman et al. [7].

This work is motivated by the following questions: *Is the $O(dk/\epsilon)$ space bound tight or can one achieve an even smaller amount of space? What if one also allows randomization?*

In this work we answer the above questions. Our main theorem is the following.

**Theorem 1.** *Any, possibly randomized, streaming algorithm in the row-update model which outputs a $k \times d$ matrix $R$ and guarantees that $\|A - AR^{\dagger}R\|_F^2 \leq (1+\epsilon)\|A - A_k\|_F^2$ with probability at least $2/3$, must use $\Omega(kd/\epsilon)$ bits of space.*

Up to a factor of the word size (which is typically $O(\log(nd))$ bits), our main theorem shows that the algorithm of Liberty is optimal. It also shows that allowing for randomization and a small probability of error does not significantly help in reducing the memory required. We note that a simple argument gives an $\Omega(kd)$ bit lower bound, see Lemma 2 below, which intuitively follows from the fact that if $A$ itself is rank-$k$, then $R$ needs to have the same rowspace of $A$, and specifying a random $k$-dimensional subspace of $\mathbb{R}^d$ requires $\Omega(kd)$ bits. Hence, the main interest here is improving upon this lower bound to $\Omega(kd/\epsilon)$ bits of space. This extra $1/\epsilon$ factor is significant for small values of $\epsilon$, e.g., if one wants approximations as close to machine precision as possible with a given amount of memory.

The only other lower bounds for streaming algorithms for low rank approximation that we know of are due to Clarkson and Woodruff [5]. As in their work, we use the Index problem in communication complexity to establish our bounds, which is a communication game between two players Alice and Bob, holding a string $x \in \{0,1\}^r$ and an index $i \in [r] =: \{1, 2, \ldots, r\}$, respectively. In this game Alice sends a single message to Bob who should output $x_i$ with constant probability. It is known (see, e.g., [13]) that this problem requires Alice's message to be $\Omega(r)$ bits long. If Alg is a streaming algorithm for low rank approximation, and Alice can create a matrix $A_x$ while Bob can create a matrix $B_i$ (depending on their respective inputs $x$ and $i$), then if from the output of Alg on the concatenated matrix $[A_x; B_i]$ Bob can output $x_i$ with constant probability, then the memory required of Alg is $\Omega(r)$ bits, since Alice's message is the state of Alg after running it on $A_x$.

The main technical challenges are thus in showing how to choose $A_x$ and $B_i$, as well as showing how the output of Alg on $[A_x; B_i]$ can be used to solve Index. This is where our work departs significantly from that of Clarkson and Woodruff [5]. Indeed, a major challenge is that in Theorem 1, we only require the output to be the matrix $R$, whereas in Clarkson and Woodruff's work from the output one can reconstruct $AR^{\dagger}R$. This causes technical complications, since there is much less information in the output of the algorithm to use to solve the communication game.

The intuition behind the proof of Theorem 1 is that given a $2 \times d$ matrix $A = [1, x; 1, 0^d]$, where $x$ is a random unit vector, then if $P = R^\dagger R$ is a sufficiently good projection matrix for the low rank approximation problem on $A$, then the second row of $AP$ actually reveals a lot of information about $x$. This may be counterintuitive at first, since one may think that $[1, 0^d; 1, 0^d]$ is a perfectly good low rank approximation. However, it turns out that $[1, x/2; 1, x/2]$ is a much better low rank approximation in Frobenius norm, and even this is not optimal. Therefore, Bob, who has $[1, 0^d]$ *together* with the output $P$, can compute the second row of $AP$, which necessarily reveals a lot of information about $x$ (e.g., if $AP \approx [1, x/2; 1, x/2]$, its second row would reveal a lot of information about $x$), and therefore one could hope to embed an instance of the Index problem into $x$. Most of the technical work is about reducing the general problem to this $2 \times d$ primitive problem.

## 2    Main Theorem

This section is devoted to proving Theorem 1. We start with a simple lemma showing an $\Omega(kd)$ lower bound, which we will refer to. The proof of this lemma is in the full version.

**Lemma 2.** *Any streaming algorithm which, for every input A, with constant probability (over its internal randomness) succeeds in outputting a matrix R for which $\|A - AR^\dagger R\|_F \le (1 + \epsilon)\|A - A_k\|_F$ must use $\Omega(kd)$ bits of space.*

Returning to the proof of Theorem 1, let $c > 0$ be a small constant to be determined. We consider the following two player problem between Alice and Bob: Alice has a $ck/\epsilon \times d$ matrix $A$ which can be written as a block matrix $[I, R]$, where $I$ is the $ck/\epsilon \times ck/\epsilon$ identity matrix, and $R$ is a $ck/\epsilon \times (d - ck/\epsilon)$ matrix in which the entries are in $\{-1/(d - ck/\epsilon)^{1/2}, +1/(d - ck/\epsilon)^{1/2}\}$. Here $[I, R]$ means we append the columns of $I$ to the left of the columns of $R$. Bob is given a set of $k$ standard unit vectors $e_{i_1}, \ldots, e_{i_k}$, for distinct $i_1, \ldots, i_k \in [ck/\epsilon] = \{1, 2, \ldots, ck/\epsilon\}$. Here we need $c/\epsilon > 1$, but we can assume $\epsilon$ is less than a sufficiently small constant, as otherwise we would just need to prove an $\Omega(kd)$ lower bound, which is established by Lemma 2.

Let $B$ be the matrix $[A; e_{i_1}, \ldots, e_{i_k}]$ obtained by stacking $A$ on top of the vectors $e_{i_1}, \ldots, e_{i_k}$. The goal is for Bob to output a rank-$k$ projection matrix $P \in \mathbb{R}^{d \times d}$ for which $\|B - BP\|_F \le (1 + \epsilon)\|B - B_k\|_F$.

Denote this problem by $f$. We will show the randomized 1-way communication complexity of this problem $R_{1/4}^{1-way}(f)$, in which Alice sends a single message to Bob and Bob fails with probability at most $1/4$, is $\Omega(kd/\epsilon)$ bits. More precisely, let $\mu$ be the following product distribution on Alice and Bob's inputs: the entries of $R$ are chosen independently and uniformly at random in $\{-1/(d - ck/\epsilon)^{1/2}, +1/(d - ck/\epsilon)^{1/2}\}$, while $\{i_1, \ldots, i_k\}$ is a uniformly random set among all sets of $k$ distinct indices in $[ck/\epsilon]$. We will show that $D_{\mu, 1/4}^{1-way}(f) = \Omega(kd/\epsilon)$, where $D_{\mu, 1/4}^{1-way}(f)$ denotes the minimum communication cost over all deterministic 1-way (from Alice to Bob) protocols which fail with probability at most $1/4$ when the inputs are distributed according to $\mu$. By Yao's minimax principle (see, e.g., [14]), $R_{1/4}^{1-way}(f) \ge D_{\mu, 1/4}^{1-way}(f)$.

We use the following two-player problem Index in order to lower bound $D_{\mu, 1/4}^{1-way}(f)$. In this problem Alice is given a string $x \in \{0, 1\}^r$, while Bob is given an index $i \in [r]$. Alice sends a single message to Bob, who needs to output $x_i$ with probability at least $2/3$. Again by Yao's minimax principle, we have that $R_{1/3}^{1-way}(\text{Index}) \ge D_{\nu, 1/3}^{1-way}(\text{Index})$, where $\nu$ is the distribution for which $x$ and $i$ are chosen independently and uniformly at random from their respective domains. The following is well-known.

**Fact 3.** *[13] $D_{\nu, 1/3}^{1-way}(\text{Index}) = \Omega(r)$.*

**Theorem 4.** *For c a small enough positive constant, and $d \ge k/\epsilon$, we have $D_{\mu, 1/4}^{1-way}(f) = \Omega(dk/\epsilon)$.*

*Proof.* We will reduce from the Index problem with $r = (ck/\epsilon)(d - ck/\epsilon)$. Alice, given her string $x$ to Index, creates the $ck/\epsilon \times d$ matrix $A = [I, R]$ as follows. The matrix $I$ is the $ck/\epsilon \times ck/\epsilon$ identity matrix, while the matrix $R$ is a $ck/\epsilon \times (d - ck/\epsilon)$ matrix with entries in $\{-1/(d - ck/\epsilon)^{1/2}, +1/(d - ck/\epsilon)^{1/2}\}$. For an arbitrary bijection between the coordinates of $x$ and the entries of $R$, Alice sets a

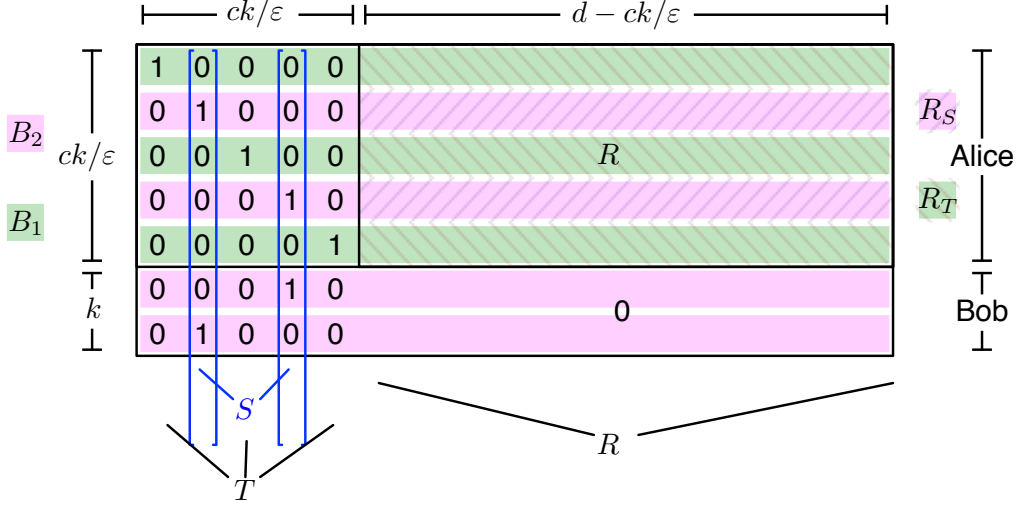

given entry in $R$ to $-1/(d-ck/\epsilon)^{1/2}$ if the corresponding coordinate of $x$ is 0, otherwise Alice sets the given entry in $R$ to $+1/(d-ck/\epsilon)^{1/2}$. In the Index problem, Bob is given an index, which under the bijection between coordinates of $x$ and entries of $R$, corresponds to being given a row index $i$ and an entry $j$ in the $i$-th row of $R$ that he needs to recover. He sets $i_\ell = i$ for a random $\ell \in [k]$, and chooses $k-1$ distinct and random indices $i_j \in [ck/\epsilon] \setminus \{i_\ell\}$, for $j \in [k] \setminus \{\ell\}$. Observe that if $(x,i) \sim \nu$, then $(R, i_1, \ldots, i_k) \sim \mu$. Suppose there is a protocol in which Alice sends a single message to Bob who solves $f$ with probability at least $3/4$ under $\mu$. We show that this can be used to solve Index with probability at least $2/3$ under $\nu$. The theorem will follow by Fact 3. Consider the matrix $B$ which is the matrix $A$ stacked on top of the rows $e_{i_1}, \ldots, e_{i_k}$, in that order, so that $B$ has $ck/\epsilon + k$ rows.

We proceed to lower bound $\|B - BP\|_F^2$ in a certain way, which will allow our reduction to Index to be carried out. We need the following fact:

**Fact 5.** *((2.4) of [20]) Let $A$ be an $m \times n$ matrix with i.i.d. entries which are each $+1/\sqrt{n}$ with probability $1/2$ and $-1/\sqrt{n}$ with probability $1/2$, and suppose $m/n < 1$. Then for all $t > 0$,*

$$\Pr[\|A\|_2 > 1 + t + \sqrt{m/n}] \leq \alpha e^{-\alpha' n t^{3/2}}.$$

*where $\alpha, \alpha' > 0$ are absolute constants. Here $\|A\|_2$ is the operator norm $\sup_x \|Ax\|/\|x\|$ of $A$.*

We apply Fact 5 to the matrix $R$, which implies,

$$\Pr[\|R\|_2 > 1 + \sqrt{c} + \sqrt{(ck/\epsilon)/(d - (ck/\epsilon))}] \leq \alpha e^{-\alpha'(d-(ck/\epsilon))c^{3/4}},$$

and using that $d \geq k/\epsilon$ and $c > 0$ is a sufficiently small constant, this implies

$$\Pr[\|R\|_2 > 1 + 3\sqrt{c}] \leq e^{-\beta d}, \tag{1}$$

where $\beta > 0$ is an absolute constant (depending on $c$). Note that for $c > 0$ sufficiently small, $(1 + 3\sqrt{c})^2 \leq 1 + 7\sqrt{c}$. Let $\mathcal{E}$ be the event that $\|R\|_2^2 \leq 1 + 7\sqrt{c}$, which we condition on.

We partition the rows of $B$ into $B_1$ and $B_2$, where $B_1$ contains those rows whose projection onto the first $ck/\epsilon$ coordinates equals $e_i$ for some $i \notin \{i_1, \ldots, i_k\}$. Note that $B_1$ is $(ck/\epsilon - k) \times d$ and $B_2$ is $2k \times d$. Here, $B_2$ is $2k \times d$ since it includes the rows in $A$ indexed by $i_1, \ldots, i_k$, together with the rows $e_{i_1}, \ldots, e_{i_k}$. Let us also partition the rows of $R$ into $R_T$ and $R_S$, so that the union of the rows in $R_T$ and in $R_S$ is equal to $R$, where the rows of $R_T$ are the rows of $R$ in $B_1$, and the rows of $R_S$ are the non-zero rows of $R$ in $B_2$ (note that $k$ of the rows are non-zero and $k$ are zero in $B_2$ restricted to the columns in $R$).

**Lemma 6.** *For any unit vector $u$, write $u = u_R + u_S + u_T$, where $S = \{i_1, \ldots, i_k\}, T = [ck/\epsilon] \setminus S$, and $R = [d] \setminus [ck/\epsilon]$, and where $u_A$ for a set $A$ is 0 on indices $j \notin A$. Then, conditioned on $\mathcal{E}$ occurring, $\|Bu\|^2 \leq (1 + 7\sqrt{c})(2 - \|u_T\|^2 - \|u_R\|^2 + 2\|u_S + u_T\|\|u_R\|)$.*

*Proof.* Let $C$ be the matrix consisting of the top $ck/\epsilon$ rows of $B$, so that $C$ has the form $[I, R]$, where $I$ is a $ck/\epsilon \times ck/\epsilon$ identity matrix. By construction of $B$, $\|Bu\|^2 = \|u_S\|^2 + \|Cu\|^2$. Now, $Cu = u_S + u_T + Ru_R$, and so

$$
\begin{aligned}
\|Cu\|_2^2 &= \|u_S + u_T\|^2 + \|Ru_R\|^2 + 2(u_s + u_T)^T Ru_R \\
&\leq \|u_S + u_T\|^2 + (1 + 7\sqrt{c})\|u_R\|^2 + 2\|u_S + u_T\|\|Ru_R\| \\
&\leq (1 + 7\sqrt{c})(\|u_S\|^2 + \|u_T\|^2 + \|u_R\|^2) + (1 + 3\sqrt{c})2\|u_S + u_T\|\|u_R\| \\
&\leq (1 + 7\sqrt{c})(1 + 2\|u_S + u_T\|\|u_R\|),
\end{aligned}
$$

and so

$$
\begin{aligned}
\|Bu\|^2 &\leq (1 + 7\sqrt{c})(1 + \|u_S\|^2 + 2\|u_S + u_T\|\|u_R\|) \\
&= (1 + 7\sqrt{c})(2 - \|u_R\|^2 - \|u_T\|^2 + 2\|u_S + U_T\|\|u_R\|).
\end{aligned}
$$

$\square$

We will also make use of the following simple but tedious fact, shown in the full version.

**Fact 7.** *For $x \in [0, 1]$, the function $f(x) = 2x\sqrt{1 - x^2} - x^2$ is maximized when $x = \sqrt{1/2 - \sqrt{5}/10}$. We define $\zeta$ to be the value of $f(x)$ at its maximum, where $\zeta = 2/\sqrt{5} + \sqrt{5}/10 - 1/2 \approx .618$.*

**Corollary 8.** *Conditioned on $\mathcal{E}$ occurring, $\|B\|_2^2 \leq (1 + 7\sqrt{c})(2 + \zeta)$.*

*Proof.* By Lemma 6, for any unit vector $u$,

$$\|Bu\|^2 \leq (1 + 7\sqrt{c})(2 - \|u_T\|^2 - \|u_R\|^2 + 2\|u_S + u_T\|\|u_R\|).$$

Suppose we replace the vector $u_S + u_T$ with an arbitrary vector supported on coordinates in $S$ with the same norm as $u_S + u_T$. Then the right hand side of this expression cannot increase, which means it is maximized when $\|u_T\| = 0$, for which it equals $(1 + 7\sqrt{c})(2 - \|u_R\|^2 + 2\sqrt{1 - \|u_R\|^2}\|u_R\|)$, and setting $\|u_R\|$ to equal the $x$ in Fact 7, we see that this expression is at most $(1 + 7\sqrt{c})(2 + \zeta)$. $\square$

Write the projection matrix $P$ output by the streaming algorithm as $UU^T$, where $U$ is $d \times k$ with orthonormal columns $u^i$ (so $R^\dagger R = P$ in the notation of Section 1). Applying Lemma 6 and Fact 7 to each of the columns $u^i$, we show in the full version:

$$
\|BP\|_F^2 \leq (1 + 7\sqrt{c})((2 + \zeta)k - \sum_{i=1}^{k} \|u_T^i\|^2). \tag{2}
$$

Using the matrix Pythagorean theorem, we thus have,

$$
\begin{aligned}
\|B - BP\|_F^2 &= \|B\|_F^2 - \|BP\|_F^2 \\
&\geq 2ck/\epsilon + k - (1 + 7\sqrt{c})((2 + \zeta)k - \sum_{i=1}^{k} \|u_T^i\|^2) \text{ using } \|B\|_F^2 = 2ck/\epsilon + k \\
&\geq 2ck/\epsilon + k - (1 + 7\sqrt{c})(2 + \zeta)k + (1 + 7\sqrt{c})\sum_{i=1}^{k} \|u_T^i\|^2. \tag{3}
\end{aligned}
$$

We now argue that $\|B - BP\|_F^2$ cannot be too large if Alice and Bob succeed in solving $f$. First, we need to upper bound $\|B - B_k\|_F^2$. To do so, we create a matrix $\tilde{B}_k$ of rank-$k$ and bound $\|B - \tilde{B}_k\|_F^2$. Matrix $\tilde{B}_k$ will be 0 on the rows in $B_1$. We can group the rows of $B_2$ into $k$ pairs so that each pair has the form $e_i + v^i, e_i$, where $i \in [ck/\epsilon]$ and $v^i$ is a unit vector supported on $[d] \setminus [ck/\epsilon]$. We let $Y_i$ be the optimal (in Frobenius norm) rank-1 approximation to the matrix $[e_i + v^i; e_i]$. By direct computation [1] the maximum squared singular value of this matrix is $2 + \zeta$. Our matrix $\tilde{B}_k$ then consists of a single $Y_i$ for each pair in $B_2$. Observe that $\tilde{B}_k$ has rank at most $k$ and

$$\|B - B_k\|_F^2 \leq \|B - \tilde{B}_k\|_F^2 \leq 2ck/\epsilon + k - (2 + \zeta)k,$$

Therefore, if Bob succeeds in solving $f$ on input $B$, then,

$$\|B - BP\|_F^2 \quad \leq \quad (1 + \epsilon)(2ck/\epsilon + k - (2 + \zeta)k) \leq 2ck/\epsilon + k - (2 + \zeta)k + 2ck. \quad (4)$$

Comparing (3) and (4), we arrive at, conditioned on $\mathcal{E}$:

$$\sum_{i=1}^{k} \|u_T^i\|^2 \leq \frac{1}{1 + 7\sqrt{c}} \cdot (7\sqrt{c}(2 + \zeta)k + 2ck) \leq c_1 k, \quad (5)$$

where $c_1 > 0$ is a constant that can be made arbitrarily small by making $c > 0$ an arbitrarily small.

Since $P$ is a projector, $\|BP\|_F = \|BU\|_F$. Write $U = \hat{U} + \bar{U}$, where the vectors in $\hat{U}$ are supported on $T$, and the vectors in $\bar{U}$ are supported on $[d] \setminus T$. We have,

$$\|B\hat{U}\|_F^2 \leq \|B\|_2^2 c_1 k \leq (1 + 7\sqrt{c})(2 + \zeta)c_1 k \leq c_2 k,$$

where the first inequality uses $\|B\hat{U}\|_F \leq \|B\|_2 \|\hat{U}\|_F$ and (5), the second inequality uses that event $\mathcal{E}$ occurs, and the third inequality holds for a constant $c_2 > 0$ that can be made arbitrarily small by making the constant $c > 0$ arbitrarily small.

Combining with (4) and using the triangle inequality,

$$
\begin{aligned}
\|B\bar{U}\|_F &\geq \|BP\|_F - \|B\hat{U}\|_F \text{ using the triangle inequality} \\
&\geq \|BP\|_F - \sqrt{c_2 k} \text{ using our bound on } \|B\hat{U}\|_F^2 \\
&= \sqrt{\|B\|_F^2 - \|B - BP\|_F^2} - \sqrt{c_2 k} \text{ by the matrix Pythagorean theorem} \\
&\geq \sqrt{(2 + \zeta)k - 2ck} - \sqrt{c_2 k} \text{ by (4)} \\
&\geq \sqrt{(2 + \zeta)k - c_3 k}, \quad (6)
\end{aligned}
$$

where $c_3 > 0$ is a constant that can be made arbitrarily small for $c > 0$ an arbitrarily small constant (note that $c_2 > 0$ also becomes arbitrarily small as $c > 0$ becomes arbitrarily small). Hence, $\|B\bar{U}\|_F^2 \geq (2 + \zeta)k - c_3 k$, and together with Corollary 8, that implies $\|\bar{U}\|_F^2 \geq k - c_4 k$ for a constant $c_4$ that can be made arbitrarily small by making $c > 0$ arbitrarily small.

Our next goal is to show that $\|B_2 \bar{U}\|_F^2$ is almost as large as $\|B\bar{U}\|_F^2$. Consider any column $\bar{u}$ of $\bar{U}$, and write it as $\bar{u}_S + \bar{u}_R$. Hence,

$$
\begin{aligned}
\|B\bar{u}\|^2 &= \|R_T \bar{u}_R\|^2 + \|B_2 \bar{u}\|^2 \text{ using } B_1 \bar{u} = R_T \bar{u}_R \\
&\leq \|R_T \bar{u}_R\|^2 + \|\bar{u}_S + R_S \bar{u}_R\|^2 + \|\bar{u}_S\|^2 \text{ by definition of the components} \\
&= \|R\bar{u}_R\|^2 + 2\|\bar{u}_S\|^2 + 2\bar{u}_S^T R_S \bar{u}_R \text{ using the Pythagorean theorem} \\
&\leq 1 + 7\sqrt{c} + \|\bar{u}_S\|^2 + 2\|\bar{u}_S\|\|R_S \bar{u}_R\| \\
&\quad \text{using } \|R\bar{u}_R\|^2 \leq (1 + 7\sqrt{c})\|\bar{u}_R\|^2 \text{ and } \|\bar{u}_R\|^2 + \|\bar{u}_S\|^2 \leq 1 \\
&\quad \text{(also using Cauchy-Schwarz to bound the other term).}
\end{aligned}
$$

Suppose $\|R_S \bar{u}_R\| = \tau \|\bar{u}_R\|$ for a value $0 \leq \tau \leq 1 + 7\sqrt{c}$. Then

$$\|B\bar{u}\|^2 \leq 1 + 7\sqrt{c} + \|\bar{u}_S\|^2 + 2\tau\|\bar{u}_S\|\sqrt{1 - \|\bar{u}_S\|^2}.$$

We thus have,

$$
\begin{aligned}
\|B\bar{u}\|^2 &\leq 1 + 7\sqrt{c} + (1 - \tau)\|\bar{u}_S\|^2 + \tau(\|\bar{u}_S\|^2 + 2\|\bar{u}_S\|\sqrt{1 - \|\bar{u}_S\|^2}) \\
&\leq 1 + 7\sqrt{c} + (1 - \tau) + \tau(1 + \zeta) \text{ by Fact 7} \\
&\leq 2 + \tau\zeta + 7\sqrt{c}, \quad (7)
\end{aligned}
$$

and hence, letting $\tau_1, \ldots, \tau_k$ denote the corresponding values of $\tau$ for the $k$ columns of $\bar{U}$, we have

$$\|B\bar{U}\|_F^2 \leq (2 + 7\sqrt{c})k + \zeta \sum_{i=1}^{k} \tau_i. \quad (8)$$

Comparing the square of (6) with (8), we have

$$\sum_{i=1}^{k} \tau_i \quad \geq \quad k - c_5 k, \quad (9)$$

where $c_5 > 0$ is a constant that can be made arbitrarily small by making $c > 0$ an arbitrarily small constant. Now, $\|\bar{U}\|_F^2 \geq k - c_4 k$ as shown above, while since $\|R_s \bar{u}_R\| = \tau_i \|\bar{u}_R\|$ if $\bar{u}_R$ is the $i$-th column of $\bar{U}$, by (9) we have

$$\|R_S \bar{U}_R\|_F^2 \geq (1 - c_6)k \tag{10}$$

for a constant $c_6$ that can be made arbitrarily small by making $c > 0$ an arbitrarily small constant.

Now $\|R\bar{U}_R\|_F^2 \leq (1 + 7\sqrt{c})k$ since event $\mathcal{E}$ occurs, and $\|R\bar{U}_R\|_F^2 = \|R_T \bar{U}_R\|_F^2 + \|R_S \bar{U}_R\|_F^2$ since the rows of $R$ are the concatenation of rows of $R_S$ and $R_T$, so combining with (10), we arrive at

$$\|R_T \bar{U}_R\|_F^2 \quad \leq \quad c_7 k, \tag{11}$$

for a constant $c_7 > 0$ that can be made arbitrarily small by making $c > 0$ arbitrarily small.

Combining the square of (6) with (11), we thus have

$$
\begin{aligned}
\|B_2 \bar{U}\|_F^2 &= \|B\bar{U}\|_F^2 - \|B_1 \bar{U}\|_F^2 = \|B\bar{U}\|_F^2 - \|R_T \bar{U}_R\|_F^2 \geq (2 + \zeta)k - c_3 k - c_7 k \\
&\geq (2 + \zeta)k - c_8 k,
\end{aligned} \tag{12}
$$

where the constant $c_8 > 0$ can be made arbitrarily small by making $c > 0$ arbitrarily small.

By the triangle inequality,

$$\|B_2 U\|_F \geq \|B_2 \bar{U}\|_F - \|B_2 \hat{U}\|_F \geq ((2 + \zeta)k - c_8 k)^{1/2} - (c_2 k)^{1/2}. \tag{13}$$

Hence,

$$
\begin{aligned}
\|B_2 - B_2 P\|_F &= \sqrt{\|B_2\|_F^2 - \|B_2 U\|_F^2} \text{ Matrix Pythagorean, } \|B_2 U\|_F = \|B_2 P\|_F \\
&\leq \sqrt{\|B_2\|_F^2 - (\|B_2 \bar{U}\|_F - \|B_2 \hat{U}\|_F)^2} \text{ Triangle Inequality} \\
&\leq \sqrt{3k - (((2 + \zeta)k - c_8 k)^{1/2} - (c_2 k)^{1/2})^2} \text{ Using (13)}, \|B_2\|_F^2 = 3k, \text{(14)}
\end{aligned}
$$
$$\tag{15}$$

or equivalently, $\|B_2 - B_2 P\|_F^2 \leq 3k - ((2 + \zeta)k - c_8 k) - (c_2 k) + 2k(((2 + \zeta) - c_8)c_2)^{1/2} \leq (1 - \zeta)k + c_8 k + 2k(((2 + \zeta) - c_8)c_2)^{1/2} \leq (1 - \zeta)k + c_9 k$ for a constant $c_9 > 0$ that can be made arbitrarily small by making the constant $c > 0$ small enough. This intuitively says that $P$ provides a good low rank approximation for the matrix $B_2$. Notice that by (14),

$$\|B_2 P\|_F^2 = \|B_2\|_F^2 - \|B_2 - B_2 P\|_F^2 \geq 3k - (1 - \zeta)k - c_9 k \geq (2 + \zeta)k - c_9 k. \tag{16}$$

Now $B_2$ is a $2k \times d$ matrix and we can partition its rows into $k$ pairs of rows of the form $Z_\ell = (e_{i_\ell} + R_{i_\ell}, e_{i_\ell})$, for $\ell = 1, \ldots, k$. Here we abuse notation and think of $R_{i_\ell}$ as a $d$-dimensional vector, its first $ck/\epsilon$ coordinates set to 0. Each such pair of rows is a rank-2 matrix, which we abuse notation and call $Z_\ell^T$. By direct computation[2] $Z_\ell^T$ has squared maximum singular value $2 + \zeta$. We would like to argue that the projection of $P$ onto the row span of most $Z_\ell$ has length very close to 1. To this end, for each $Z_\ell$ consider the orthonormal basis $V_\ell^T$ of right singular vectors for its row space (which is $\text{span}(e_{i_\ell}, R_{i_\ell})$). We let $v_{\ell,1}^T, v_{\ell,2}^T$ be these two right singular vectors with corresponding singular values $\sigma_1$ and $\sigma_2$ (which will be the same for all $\ell$, see below). We are interested in the quantity $\Delta = \sum_{\ell=1}^k \|V_\ell^T P\|_F^2$ which intuitively measures how much of $P$ gets projected onto the row spaces of the $Z_\ell^T$. The following lemma and corollary are shown in the full version.

**Lemma 9.** *Conditioned on event $\mathcal{E}$, $\Delta \in [k - c_{10}k, k + c_{10}k]$, where $c_{10} > 0$ is a constant that can be made arbitrarily small by making $c > 0$ arbitrarily small.*

The following corollary is shown in the full version.

**Corollary 10.** *Conditioned on event $\mathcal{E}$, for a $1 - \sqrt{c_9 + 2c_{10}}$ fraction of $\ell \in [k]$, $\|V_\ell^T P\|_F^2 \leq 1 + c_{11}$, and for a $99/100$ fraction of $\ell \in [k]$, we have $\|V_\ell^T P\|_F^2 \geq 1 - c_{11}$, where $c_{11} > 0$ is a constant that can be made arbitrarily small by making the constant $c > 0$ arbitrarily small.*

Recall that Bob holds $i = i_\ell$ for a random $\ell \in [k]$. It follows (conditioned on $\mathcal{E}$) by a union bound that with probability at least $49/50$, $\|V_\ell^T P\|_F^2 \in [1 - c_{11}, 1 + c_{11}]$, which we call the event $\mathcal{F}$ and condition on. We also condition on event $\mathcal{G}$ that $\|Z_\ell^T P\|_F^2 \geq (2+\zeta) - c_{12}$, for a constant $c_{12} > 0$ that can be made arbitrarily small by making $c > 0$ an arbitrarily small constant. Combining the first part of Corollary 10 together with (16), event $\mathcal{G}$ holds with probability at least $99.5/100$, provided $c > 0$ is a sufficiently small constant. By a union bound it follows that $\mathcal{E}, \mathcal{F}$, and $\mathcal{G}$ occur simultaneously with probability at least $49/51$.

As $\|Z_\ell^T P\|_F^2 = \sigma_1^2 \|v_{\ell,1}^T P\|^2 + \sigma_2^2 \|v_{\ell,2}^T P\|^2$, with $\sigma_1^2 = 2 + \zeta$ and $\sigma_1^2 = 1 - \zeta$, events $\mathcal{E}, \mathcal{F}$, and $\mathcal{G}$ imply that $\|v_{\ell,1}^T P\|^2 \geq 1 - c_{13}$, where $c_{13} > 0$ is a constant that can be made arbitrarily small by making the constant $c > 0$ arbitrarily small. Observe that $\|v_{\ell,1}^T P\|^2 = \langle v_{\ell,1}, z \rangle^2$, where $z$ is a unit vector in the direction of the projection of $v_{\ell,1}$ onto $P$.

By the Pythagorean theorem, $\|v_{\ell,1} - \langle v_{\ell,1}, z \rangle z\|^2 = 1 - \langle v_{\ell,1}, z \rangle^2$, and so

$$\|v_{\ell,1} - \langle v_{\ell,1}, z \rangle z\|^2 \leq c_{14}, \tag{17}$$

for a constant $c_{14} > 0$ that can be made arbitrarily small by making $c > 0$ arbitrarily small.

We thus have $Z_\ell^T P = \sigma_1 \langle v_{\ell,1}, z \rangle u_{\ell,1} z^T + \sigma_2 \langle v_{\ell,2}, w \rangle u_{\ell,2} w^T$, where $w$ is a unit vector in the direction of the projection of of $v_{\ell,2}$ onto $P$, and $u_{\ell,1}, u_{\ell,2}$ are the left singular vectors of $Z_\ell^T$. Since $\mathcal{F}$ occurs, we have that $|\langle v_{\ell,2}, w \rangle| \leq c_{11}$, where $c_{11} > 0$ is a constant that can be made arbitrarily small by making the constant $c > 0$ arbitrarily small. It follows now by (17) that

$$\|Z_\ell^T P - \sigma_1 u_{\ell,1} v_{\ell,1}^t\|_F^2 \leq c_{15}, \tag{18}$$

where $c_{15} > 0$ is a constant that can be made arbitrarily small by making the constant $c > 0$ arbitrarily small.

By direct calculation[3] , $u_{\ell,1} = -.851 e_{i_\ell} - .526 R_{i_\ell}$ and $v_{\ell,1} = -.851 e_{i_\ell} - .526 R_{i_\ell}$. It follows that $\|Z_\ell^T P - (2 + \zeta)[.724 e_{i_\ell} + .448 R_{i_\ell}; .448 e_{i_\ell} + .277 R_{i_\ell}]\|_F^2 \leq c_{15}$. Since $e_{i_\ell}$ is the second row of $Z_\ell^T$, it follows that $\|e_{i_\ell}^T P - (2 + \zeta)(.448 e_{i_\ell} + .277 R_{i_\ell})\|^2 \leq c_{15}$.

Observe that Bob has $e_{i_\ell}$ and $P$, and can therefore compute $e_{i_\ell}^T P$. Moreover, as $c_{15} > 0$ can be made arbitrarily small by making the constant $c > 0$ arbitrarily small, it follows that a $1 - c_{16}$ fraction of the signs of coordinates of $e_{i_\ell}^T P$, restricted to coordinates in $[d] \setminus [ck/\epsilon]$, must agree with those of $(2 + \zeta).277 R_{i_\ell}$, which in turn agree with those of $R_{i_\ell}$. Here $c_{16} > 0$ is a constant that can be made arbitrarily small by making the constant $c > 0$ arbitrarily small. Hence, in particular, the sign of the $j$-th coordinate of $R_{i_\ell}$, which Bob needs to output, agrees with that of the $j$-th coordinate of $e_{i_\ell}^T P$ with probability at least $1 - c_{16}$. Call this event $\mathcal{H}$.

By a union bound over the occurrence of events $\mathcal{E}, \mathcal{F}, \mathcal{G}$, and $\mathcal{H}$, and the streaming algorithm succeeding (which occurs with probability $3/4$), it follows that Bob succeeds in solving Index with probability at least $49/51 - 1/4 - c_{16} > 2/3$, as required. This completes the proof. $\qquad\square$

## 3   Conclusion

We have shown an $\Omega(dk/\epsilon)$ bit lower bound for streaming algorithms in the row-update model for outputting a $k \times d$ matrix $R$ with $\|A - AR^\dagger R\|_F \leq (1 + \epsilon)\|A - A_k\|_F$, thus showing that the algorithm of [9] is optimal up to the word size. The next natural goal would be to obtain multi-pass lower bounds, which seem quite challenging. Such lower bound techniques may also be useful for showing the optimality of a constant-round $O(sdk/\epsilon) + (sk/\epsilon)^{O(1)}$ communication protocol in [12] for low-rank approximation in the distributed communication model.

**Acknowledgments.**   I would like to thank Edo Liberty and Jeff Phillips for many useful discussions and detailed comments on this work (thanks to Jeff for the figure!). I would also like to thank the XDATA program of the Defense Advanced Research Projects Agency (DARPA), administered through Air Force Research Laboratory contract FA8750-12-C0323 for supporting this work.

## Footnotes

[1] For an online SVD calculator, see http://www.bluebit.gr/matrix-calculator/

[2]We again used the calculator at `http://www.bluebit.gr/matrix-calculator/`

[3]Using the online calculator in earlier footnotes.

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
