[Supplementary Material]

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

# A    Proof of Lemma 2

Let $\mathcal{S}$ be the set of $k$-dimensional subspaces over the vector space $GF(2)^d$, where $GF(2)$ denotes the finite field of 2 elements with the usual modulo 2 arithmetic. The cardinality of $\mathcal{S}$ is known [18] to be

$$\frac{(2^d - 1)(2^d - 1) \cdots (2^{d-k+1} - 1)}{(2^k - 1)(2^{k-1} - 1) \cdots 1} \geq 2^{dk/2 - k^2} \geq 2^{dk/6},$$

where the inequalities assume that $k \leq d/3$.

Now let $A^1$ and $A^2$ be two $k \times d$ matrices with entries in $\{0, 1\}$ whose rows span two different $k$-dimensional subspaces of $GF(2)^d$. We first claim that the rows also span two different $k$-dimensional subspaces of $\mathbb{R}^d$. Indeed, consider a vector $v \in GF(2)^d$ which is in the span of the rows of $A^1$ but not in the span of the rows of $A^2$. If $A^1$ and $A^2$ had the same row span over $\mathbb{R}^d$, then $v = \sum_i w_i A_i^2$, where the $w_i \in \mathbb{R}$ and $A_i^2$ denotes the $i$-th row of $A^2$. Since $v$ has integer coordinates and the $A_i^2$ have integer coordinates, we can assume the $w_i$ are rational, since the irrational parts must cancel. By scaling by the least common multiple of the denominators of the $w_i$, we obtain that

$$\alpha \cdot v = \sum_i \beta_i A_i^2, \tag{19}$$

where $\alpha, \beta_1, \ldots, \beta_k$ are integers. We can assume that the greatest common divisor (gcd) of $\alpha, \beta_1, \ldots, \beta_k$ is 1, otherwise the same conclusion holds after we divide $\alpha, \beta_1, \ldots, \beta_k$ by the gcd. Note that (19) implies that $\alpha v = \sum_i \beta_i A_i^2 \mod 2$, i.e., when we take each of the coordinates modulo 2. Since the $\beta_i$ cannot all be divisible by 2 (since $\alpha$ would then be odd and so by the gcd condition the left hand side would contain a vector with at least one odd coordinate, contradicting that the right hand side is a vector with even coordinates), and the rows of $A^2$ form a basis over $GF(2^d)$, the right hand side must be non-zero, which implies that $\alpha = 1 \mod 2$. This implies that $v$ is in the span of the rows of $A^2$ over $GF(2^d)$, a contradiction.

It follows that there are at least $2^{dk/6}$ distinct $k$-dimensional subspaces of $\mathbb{R}^d$ spanned by the rows of the set of binary $k \times d$ matrices $A$. For each such $A$, $\|A - A_k\|_F = 0$ and so the row span of $R$ must agree with the row span of $A$ if the streaming algorithm succeeds. It follows that the output of the streaming algorithm can be used to encode $\log_2 2^{dk/6} = \Omega(dk)$ bits of information. Indeed, if $A$ is chosen at random from this set of at least $2^{dk/6}$ binary matrices, and $Z$ is a bit indicating if the streaming algorithm succeeds, then

$$|R| \geq H(R) \geq I(R; A|Z) \geq (2/3)I(R; A \mid Z = 1) \geq (2/3)(dk/6) = \Omega(dk),$$

where $|R|$ denotes the expected length of the encoding of $R$, $H$ is the entropy function, and $I$ is the mutual information. For background on information theory, see [7]. This completes the proof.

# B    Proof of Fact 7

Setting $y^2 = 1 - x^2$, we can equivalently maximize $f(y) = -1 + 2y\sqrt{1 - y^2} + y^2$, or equivalently $g(y) = 2y\sqrt{1 - y^2} + y^2$. Differentiating this expression and equating to 0, we have

$$2\sqrt{1 - y^2} - \frac{2y^2}{\sqrt{1 - y^2}} + 2y = 0.$$

Multiplying both sides by $\sqrt{1 - y^2}$ one obtains the equation $4y^2 - 2 = 2y\sqrt{1 - y^2}$, and squaring both sides, after some algebra one obtains $5y^4 - 5y^2 + 1 = 0$. Using the quadratic formula, we get that the maximizer satisfies $y^2 = 1/2 + \sqrt{5}/10$, or $x^2 = 1/2 - \sqrt{5}/10$.

# C   Deriviation of Inequality (2)

$$
\begin{aligned}
\|BP\|_F^2 &= \|BU\|_F^2 = \sum_{i=1}^{k} \|Bu^i\|^2 \\
&\leq (1 + 7\sqrt{c}) \sum_{i=1}^{k} (2 - \|u_T^i\|^2 - \|u_R^i\|^2 + 2\|u_S^i + u_T^i\|\|u_R^i\|) \\
&= (1 + 7\sqrt{c})(2k - \sum_{i=1}^{k}(\|u_T^i\|^2) + \sum_{i=1}^{k}(2\|u_S^i + u_T^i\|\|u_R^i\| - \|u_R^i\|^2)) \\
&= (1 + 7\sqrt{c})(2k - \sum_{i=1}^{k}(\|u_T^i\|^2) + \sum_{i=1}^{k}(2\sqrt{1 - \|u_R^i\|^2}\|u_R^i\| - \|u_R^i\|^2)) \\
&\leq (1 + 7\sqrt{c})(2k - \sum_{i=1}^{k}(\|u_T^i\|^2) + k\zeta) \\
&= (1 + 7\sqrt{c})((2 + \zeta)k - \sum_{i=1}^{k} \|u_T^i\|^2).
\end{aligned}
$$

# D   Proof of Lemma 9

For any unit vector $u$, consider $\sum_{\ell=1}^{k} \|V_\ell^T u\|^2$. This is equal to $\|u_S\|^2 + \|R_S u_R\|^2$. Conditioned on $\mathcal{E}$, $\|R_S u_R\|^2 \leq (1 + 7\sqrt{c})\|u_R\|^2$. Hence, $\sum_{\ell=1}^{k} \|V_\ell^T u\|^2 \leq 1 + 7\sqrt{c}$, and consequently, $\Delta \leq k(1 + 7\sqrt{c})$.

On the other hand, $\|B_2 P\|_F^2 = \sum_{\ell=1}^{k} \|Z_\ell^T P\|_F^2$. Since $\|Z_\ell^T\|_2^2 \leq 2 + \zeta$, it follows by (16) that $\Delta \geq k - (c_9/(2 + \zeta))k$, as otherwise $\Delta$ would be too small in order for (16) to hold.

The lemma now follows since $\sqrt{c}$ and $c_9$ can be made arbitrarily small by making the constant $c > 0$ small enough.

# E   Proof of Corollary 10

For the first part of the corollary, observe that

$$
\|Z_\ell^T P\|_F^2 = \sigma_1^2 \|v_{\ell,1}^T P\|^2 + \sigma_2^2 \|v_{\ell,2}^T P\|^2,
$$

where $v_{\ell,1}^T$ and $v_{\ell,2}^T$ are right singular vectors of $V_\ell^T$, and $\sigma_1, \sigma_2$ are its singular values, with $\sigma_1^2 = 2 + \zeta$ and $\sigma_2^2 = 1 - \zeta$. Since $\Delta \leq k + c_{10}k$ by Lemma 9, we have

$$
\sum_{\ell=1}^{k} \|v_{\ell,1}^T P\|^2 + \|v_{\ell,2}^T P\|^2 \leq k + c_{10}k.
$$

If $\sum_{\ell=1}^{k} \|v_{\ell,2}^T P\|^2 \geq (c_9 + 2c_{10})k$, then

$$
\begin{aligned}
\|B_2 P\|_F^2 &\leq \sum_{\ell} \|Z_\ell^T P\|_F^2 \\
&\leq (2 + \zeta)(k + c_{10}k - 2c_{10}k - c_9 k) + (1 - \zeta)(2c_{10}k + c_9 k) \\
&\leq (2 + \zeta)(k - c_9 k) - (2 + \zeta)c_{10}k + (1 - \zeta)(2c_{10}k + c_9 k) \\
&\leq (2 + \zeta)k - 2c_9 k - \zeta c_9 k - 2c_{10}k - \zeta c_{10}k + 2c_{10}k + c_9 k - 2\zeta c_{10}k - \zeta c_9 k \\
&\leq (2 + \zeta)k - (1 + 2\zeta)c_9 k + -3\zeta c_{10}k \\
&< (2 + \zeta)k - c_9 k
\end{aligned}
$$

which is a contradiction to (16). Hence, $\sum_{\ell=1}^{k} \|v_{\ell,2}^T P\|^2 \le (c_9 + 2c_{10})k$. This means by a Markov bound that a $1 - \sqrt{c_9 + 2c_{10}}$ fraction of $\ell$ satisfy $\|v_{\ell,2}^T P\|^2 \le \sqrt{c_9 + 2c_{10}}$, which implies that for this fraction that $\|V_\ell^T P\|_F^2 \le 1 + \sqrt{c_9 + 2c_{10}}$.

For the second part of the corollary, suppose at most $99k/100$ different $\ell$ satisfy $\|V_\ell^T P\|_F^2 > 1 - 200\sqrt{c_9 + 2c_{10}}$. By the previous part of the corollary, at most $\sqrt{c_9 + 2c_{10}}k$ of these $\ell$ can satisfy $\|V_\ell^T P\|_F^2 > 1 + \sqrt{c_9 + 2c_{10}}$. Hence, since $\|V_\ell^T P\|_F^2 \le 2$,

$$
\begin{aligned}
\Delta \quad &< \quad 2\sqrt{c_9 + 2c_{10}}k + (1 + \sqrt{c_9 + 2c_{10}})(99/100 - \sqrt{c_9 + 2c_{10}})k + (1 - 200\sqrt{c_9 + 2c_{10}})k/100 \\
&\le \quad 2\sqrt{c_9 + 2c_{10}}k + 99k/100 + 99k\sqrt{c_9 + 2c_{10}}/100 - k\sqrt{c_9 + 2c_{10}} + k/100 - 2\sqrt{c_9 + 2c_{10}}k \\
&\le \quad k - \sqrt{c_9 + 2c_{10}}k/100 \\
&\le \quad k - \sqrt{2c_{10}}k/100 \\
&< \quad k - c_{10}k,
\end{aligned}
$$

where the final inequality follows for $c_{10} > 0$ a sufficiently small constant. This is a contradiction to Lemma 9. Hence, at least $99k/100$ different $\ell$ satisfy $\|V_\ell^T P\|_F^2 > 1 - 200\sqrt{c_9 + 2c_{10}}$. Letting $c_{11} = 200\sqrt{c_9 + 2c_{10}}$, we see that $c_{11}$ can be made an arbitrarily small constant by making the constant $c > 0$ arbitrarily small. This completes the proof.