[Reviews · NeurIPS 2014]

Submitted by Assigned_Reviewer_5

Comments:
The paper is well written and it appears like a novel result, perhaps less so to someone who is an expert in streaming algorithms. The theory and Theorem 4 in particular appear to be correct.
Summary: Summary:
This paper deals with low-rank "k" approximation of a matrix when its rows are presented in a streaming fashion. Recently (SODA 2014), an algorithm was proposed for this task which uses O(dk/epsilon) words of space. It was an open question if that bound is tight. This paper shows a \Omega(dk/epsilon) lower bound which establishes that the bound is optimal and that it needs \Theta(dk/epsilon) words of space.

Submitted by Assigned_Reviewer_30

This paper investigates the lower bound of the space complexity for a specific streaming algorithm. In this streaming algorithm, the goal is to find a matrix R such that the matrix A can be approximated by A*R^\dag*R, where R^\dag is the pseudo-inverse of the matrix R. Also in the streaming model the rows of the matrix A are presented at a time in an arbitrary order.
The problem investigated in this paper is interesting. This paper gives a lower bound of space needed to compute the matrix R. Compared with existing work, the improvement of the space lower bound is impressive.
Since the main result of this paper is the improved lower bound, it is a very theoretical paper. It is not very easy for me to follow the whole paper. Many proofs in this paper are provided in the supplementary file. I went through this paper but did not check all details of this paper. But the proof of this paper sounds good to me.
In terms of quality and significance, I think this paper is above the borderline of NIPS.
My only concern is that many proofs are provided in the supplementary file, which makes reading this paper not easy. The readability can be further improved.
Minor comment: the proof of Lemma 6 can be revised. For example, "expanding the square" can be removed from the long equation.
Summary: It is a good theoretical paper which improves the lower bound of the space complexity for the low-rank approximation problem in the streaming scenario. Readability can be further improved as many proofs are in the supplementary file.

Submitted by Assigned_Reviewer_44

The concern is to compute a k-rank approximation of a $n\times d$ matrix A when its rows are presented in a streaming fashion, one at a time.

Specifically, the algorithm of Ghashami and Phillips (SODA, 2014) is shown optimal (up to word size) with a space lower bound of $\Omega(dk/e)$.

Theoretical results are presented as a reduction to the well-known Index problem in communication complexity. The exposition that follows is well written and technically sound. Once the problem is formulated in the authors' framework, the mathematical machinery to proof the statements is standard.

My major concern with the paper is the significance of the contributed results, specially when comparing with the already available in the aforementioned SODA paper.
Summary: In this paper, the O(dk/e) space requirement for the streaming low-rank matrix approximation algorithm of Ghashami and Phillips is shown tight. The paper is well written and seems technically sound.
Author Feedback
Author rebuttal: We would like to thank the reviewers for their feedback!

Regarding the first review, we can make some derivations in Lemma 6 and elsewhere more readable.

Regarding the second review, we're not sure how to compare this paper to the result in the SODA paper, as that paper has only the algorithm. We'd like to view this paper and the SODA paper as complementary, this one showing the algorithm presented in the SODA paper is optimal by presenting a matching lower bound.

Regarding the third review, we looked at known streaming lower bounds [5], which don't seem to apply because there is much less information in the output in the problem here than in the output in the problems in [5]. As observed in [10], the problem considered here may be more natural, where we just want an approximation to the top k principal components rather than an entire factorization of A in the stream. We have outlined the technical challenges faced by the low information output in our problem in the last several paragraphs of the introduction, but can expand upon them for further clarity.